# Diabetes self-management education interventions in the WHO African Region: A scoping review

Emmanuel Kumah[1]*, Godfred Otchere[2,3], Samuel Egyakwa Ankomah[4],
Adam Fusheini[4,5], Collins Kokuro[6], Kofi Aduo-Adjei[7], Joseph A. Amankwah[8]

1 Department of Health Administration and Education, Faculty of Science Education, University of Education, Winneba, Ghana, 2 Policy, Planning, Monitoring and Evaluation Unit, Komfo Anokye Teaching Hospital, Kumasi, Ghana, 3 Faculty of Humanities, Center for Medicine and Society, University of Freiburg, Freiburg im Breisgau, Germany, 4 Department of Preventive and Social Medicine, Dunedin School of Medicine, University of Otago, Dunedin, New Zealand, 5 Center for Health Literacy and Rural Health Promotion, Accra, Ghana, 6 Department of Medicine, School of Medicine and Dentistry, Kwame Nkrumah University of Science and Technology, Kumasi, Ghana, 7 Center for Health Services Management, Faculty of Health, University of Technology Sydney, Sydney, Australia, 8 Department of Administration, Ankaase Methodist Hospital, Ankaase, Ghana

* emmanuelkumah@uew.edu.gh, ababiohemmanuel@gmail.com

## Abstract

### Introduction

Diabetes mellitus (DM) is one of the commonest chronic diseases worldwide. Self-Management Education (SME) is regarded as a critical element of treatment for all people with diabetes, as well as those at risk of developing the condition. While a great variety of diabetes self-management education (DSME) interventions are available in high-income countries, limited information exists on educational programs for the prevention and management of diabetes complications in Africa. This study, therefore, aimed at synthesizing information in the literature to describe the state of the science of DSME interventions in the WHO African Region.

### Materials and methods

The study is a scoping review, which followed the standard PRISMA guidelines for conducting and reporting scoping reviews. A systematic keyword and subject headings searches were conducted on six electronic databases (PubMed, Scopus, MEDLINE, EMBASE, PsychINFO and the Cochrane Central Register of Controlled Trials) to identify relevant English language publications on DSME from 2000 through 2020. Titles and abstracts of the search results were screened to select eligible papers for full text reading. All eligible papers were retrieved and full text screening was done by three independent reviewers to select studies for inclusion in the final analysis.

### Results

Nineteen studies were included in the review. The interventions identified were individually oriented, group-based, individually oriented & group-based, and information technology-

**Data Availability Statement:** All data have been included in the paper.

**Funding:** The author(s) received no specific funding for this work.

**Competing interests:** The authors have declared that no competing interests exist.

based DSME programs. Outcomes of the interventions were mixed. While the majority yielded significant positive results on HbA1c, diabetes knowledge, blood pressure, blood sugar and foot care practices; few demonstrated positive outcomes on self-efficacy, BMI, physical activity; self-monitoring of blood glucose, medication adherence, smoking and alcohol consumption.

## Conclusions

The limited studies available indicate that DSME interventions in the WHO African Region have mixed effects on patient behaviors and health outcomes. That notwithstanding, the majority of the interventions demonstrated statistically significant positive effects on HbA1c, the main outcome measure in most DSME intervention studies.

## Introduction

Diabetes mellitus (DM) is one of the commonest chronic diseases worldwide [1, 2]. It is among the ten leading causes of mortality in adults, and was estimated to have accounted for four million deaths globally in 2017 [3]. It still continues to be the biggest endocrine driver for the Global Burden of Disease (GBD) [4]. The World Health Organization (WHO) Global Report on Diabetes indicates that the number of adults living with diabetes increased from 180 million in 1980 to 422 million in 2014—an increase in prevalence of 80.9% [5, 6]. In 2019, the International Diabetes Federation (IDF) estimated that the global diabetes prevalence was 9.3% (463 million people). This figure has been projected to rise to 10.2% (578 million) and 10.9% (700 million) by 2030 and 2045 respectively [7].

It is estimated that about 80% of people with diabetes live in low and middle-income countries [2]. Africa, which has a high proportion of the world's least developed countries, is among the continents with rapidly increasing prevalence of diabetes. For instance, diabetes prevalence in the WHO African Region increased by 129% from 4.7% in 1980 to 8.5% in 2014. This increase was second only to the WHO Eastern Mediterranean Region where the prevalence rose by 132.2% between 1980 and 2014 [5]. African countries face a significant rise in healthcare expenditure due to the increasing prevalence of diabetes. A 2017 report by the *Lancet Diabetes & Endocrinology* Commission on Diabetes in Sub-Saharan Africa (SSA) estimated that, in 2015, the overall cost of diabetes in SSA was US $19.45 billion, and this has been projected to rise to between $35.33 billion and $59.32 billion by 2030 [8].

Despite the growing burden of diabetes, available evidence indicates that its care and control are far from optimal. This has been attributed largely to the complex nature of its management, lack of adequate healthcare resources and low income levels of individuals, particularly, those in low and middle-income country settings [9].

Self-management education (SME) is regarded as a critical element of treatment for all people with diabetes, as well as those at risk of developing the condition [3, 10–13]. SME is "the process of teaching persons with chronic disease to manage their illness and treatment by providing them with the knowledge and skills that are needed to perform self-care behaviors, manage crises, and make lifestyle changes" [14]. Promoting self-management through education is in line with WHO's best practice strategy for chronic conditions, which is to "educate and support patients to manage their conditions as much as possible" [15]. Educational programs involve a variety of psychological and behavioral interventions; as well as a combination

of didactic, interactive and collaborative teaching methods tailored to patient's specific needs [16]. Content of education could be general (applicable to several chronic conditions) or specific to a condition (e.g., diabetes mellitus, bronchial asthma, systemic hypertension, etc.). Educational sessions may be held in health care settings, in the community, or at home. Delivery mode may include individual, group, or self-mediated, and may be led by lay leaders, physicians, dietitians, nurses, or other specialists [17]. Subjects covered in educational programs include: relaxation and fatigue symptom management, problem solving; managing depression; making informed treatment decisions; managing medication; cognitive skills; anger, fear and frustration management; communication skills; the role of healthy eating and exercise; planning for the future and making an action plan; and working in partnership with health care providers [18].

While a great variety of diabetes self-management education (DSME) interventions are available in high-income countries [11], limited information exists on educational programs for the prevention and management of diabetes complications in Africa, particularly, countries in the WHO African Region [19, 20]. According to Dube and colleagues, DSME in most African countries are limited in scope, content and consistency and it is unclear as to how patients from SSA manage their diabetes [20]. Another study [21] adds that there is paucity of information on the outcomes of DSME interventions in Africa. Although a 2018 systematic review to describe the level of self-management among people living with type 2 DM in SSA found that the provision of structured DSME was effective in improving patients' behaviors and health outcomes [19], the finding was based on limited data (only six out of the 43 reviewed studies were based on DSME interventions).

The aim of the present study, therefore, was to synthesize information in the extant literature to describe the state of the science of DSME interventions in the WHO African Region. We sought to determine: 1) the types of DSME interventions that have been developed and implemented in the WHO African Region; and 2) the effects of these DSME interventions on patients' behaviors and health outcomes.

## Methods

We used a scoping review, guided by the PRISMA statement for reporting scoping reviews (S1 Appendix), to gather and summarize the existing literature on DSME interventions in the WHO African Region. Our definition of DSME intervention was based on the American Association of Diabetes Educators' (AADEs') National Standards for Diabetes Self-Management Education and Support. That is, a program to "facilitate the development of knowledge, skills and abilities that are required for successful self-management of diabetes") [22].

### Search strategy

The search strategy for this review was first drafted for pre-testing in Embase (via Ovid). Once the Embase strategy was pre-tested and finalized, it was adapted to the syntax and subject headings of all of the other databases searched in the study. Keywords used in the search were "diabetes mellitus", "self-management education", "WHO African Region", "Sub-Saharan Africa". As an example, the search strategy for Embase has been included as a supplementary file (see S2 Appendix).

The search was conducted in May, 2020. The following databases and search engines were searched: PubMed, Scopus, MEDLINE, EMBASE, PsychINFO and the Cochrane Central Register of Controlled Trials. In addition, reference lists of all eligible articles identified were searched and screened for additional relevant studies. Further, we searched the grey literature for relevant unpublished studies on DSME. We restricted the search to only English language

medical literature published between January, 2000 and April 30, 2020. This date range was chosen because our aim was to review the more recent publications on DSME interventions within the WHO African Region.

## Inclusion/exclusion criteria

Studies were reviewed against pre-determined inclusion and exclusion criteria for eligibility in the final analysis. To be eligible for inclusion:

- The primary focus of the study should be on self-management education for diabetes patients

- Participants of the study should be people living with either type 1 or type 2 diabetes

- The study should evaluate the effect of a DSME intervention on patient behaviors and health outcomes

- The setting of the study should be a country from the WHO African Region as listed by the World Health Organization (https://www.who.int/choice/demography)

- The study should be an English language article published after December, 1999

    Studies were excluded if:

- The primary focus was on diabetes and other chronic conditions such as hypertension, asthma, etc.

- The primary focus was on self-management education for diabetes, but outside the WHO African Region

- They lacked outcome assessment of program effectiveness

- They were published before 2000

- They compared two or more DSME interventions (e.g., group vs. individual education) with no controls (i.e. patients with no education)

- They were review articles, editorials or qualitative studies

    Although diabetes may be associated with one or more comorbidities, studies that covered recruitment of participants with different chronic diseases with diabetes self-management not being the main focus (i.e. some having diabetes and others having hypertension, arthritis, cardiovascular disease, etc) were excluded. This was necessary as the focus of this study was on DSME and not the generic chronic disease self-management education (CDSME).

    Furthermore, since the study was interested in interventions generally and not papers comparing types of interventions, studies comparing one or two DSME interventions were also excluded. It is difficult to determine intervention effectiveness when studies compare only types of interventions (e.g. Group Education vs. Individual Education, or Individual Education vs. Group & Individual Education) with no controls (i.e. patients with no educational intervention). Thus, our decision to exclude such studies as the study also sought to assess the effectiveness of the DSME interventions.

## Study selection

Selection and inclusion of papers for this review involved a two-stage process: screening of abstracts and titles; and full text reading to select eligible papers for final inclusion. Three independent reviewers (EK, SEA and GO) conducted the selection process through each stage of

the review. All publications retrieved through the search were imported into a shared bibliography for duplicate records to be removed. After removing the duplicates, the reviewers applied the pre-determined inclusion and exclusion criteria and independently assessed the titles and abstracts for full-text review eligibility. Following this process, articles were selected for full-text screening. Again, the reviewers applied the inclusion and exclusion criteria and independently assessed the full-text articles. After each stage of the selection process, the reviewers compared results and reached a consensus, with a fourth reviewer (AF) serving as a tiebreaker in situations where the three reviewers failed to reach an agreement.

## Data extraction, analysis and synthesis

Data from the eligible papers were extracted by two members (EK, and GO) of the research team working independently, and checked by two other members (AF and CK) to ensure consistency and accuracy of the extracted information. All differences were discussed by the assessors until a consensus was reached.

Three data extraction templates were developed, using Microsoft Excel, to collect the relevant data for analysis. One template was used to collect information on characteristics of the included studies, such as: name of author, country of study, study design, purpose and study sample. Characteristics of the DSME interventions evaluated in the included studies were collected in the second template. The final template was used to gather information on the main outcomes of the DSME interventions.

A modified version of Mulcahy and colleagues' diabetes SME continuum of outcomes categories [23] was used to synthesize the outcomes reported by the included studies into three categories of outcome measures: 1) learning/immediate outcomes (e.g. knowledge acquisition, skills acquisition, self-efficacy, etc.), 2) behavioral/intermediate outcomes (changes in dietary practices, physical exercise, self-monitoring of blood glucose, medication adherence, etc.), and 3) clinical/ post intermediate outcomes (changes in glycated hemoglobin, body mass index/ weight, blood pressure, fasting lipids, fasting blood sugar, waist circumference, etc.).

## Assessment of study quality

The Effective Public Health Practice Project Quality Assessment Tool (EPHPP) (http://www.city.hamilton.on.ca/phcs/EPHPP/) (see S3 Appendix) was adopted to assess the methodological quality of the included studies. Two independent reviewers (EK and KAA) conducted the quality assessment. Each article was rated on the EPHPP six domains as strong (3 points), moderate (2 points) or weak (1 point). Domain scores were then averaged to produce total scores, with the maximum total score per study being 3.00. Based on the total scores, studies were assigned an overall quality rating of strong (2.51–3.00), moderate (1.51–2.50) or weak (1.00–1.50) as recommended by the EPHPP guidelines [24]. After completing the quality assessment of each paper, the assessors met to discuss and resolve discrepancies. Studies were not excluded on the basis of poor methodological quality.

## Results

### Literature search

The search identified a total of 3,264 papers: 3,257 from electronic database search, and 7 from manual search. Following the removal of duplicates, 2,837 articles remained. The abstracts and titles screening resulted in the exclusion of 2,649 articles, leaving 188 for full-text screening. One hundred and sixty-nine (169) articles were further excluded after the full text reading. The most common reason for paper exclusion was lack of outcome assessment of program

effectiveness (n = 73), followed by a study focusing on more than one chronic disease (n = 32), and study participants being health professionals (n = 29). In all, 19 articles were included in the final analysis. Fig 1 depicts stages of study identification and selection.

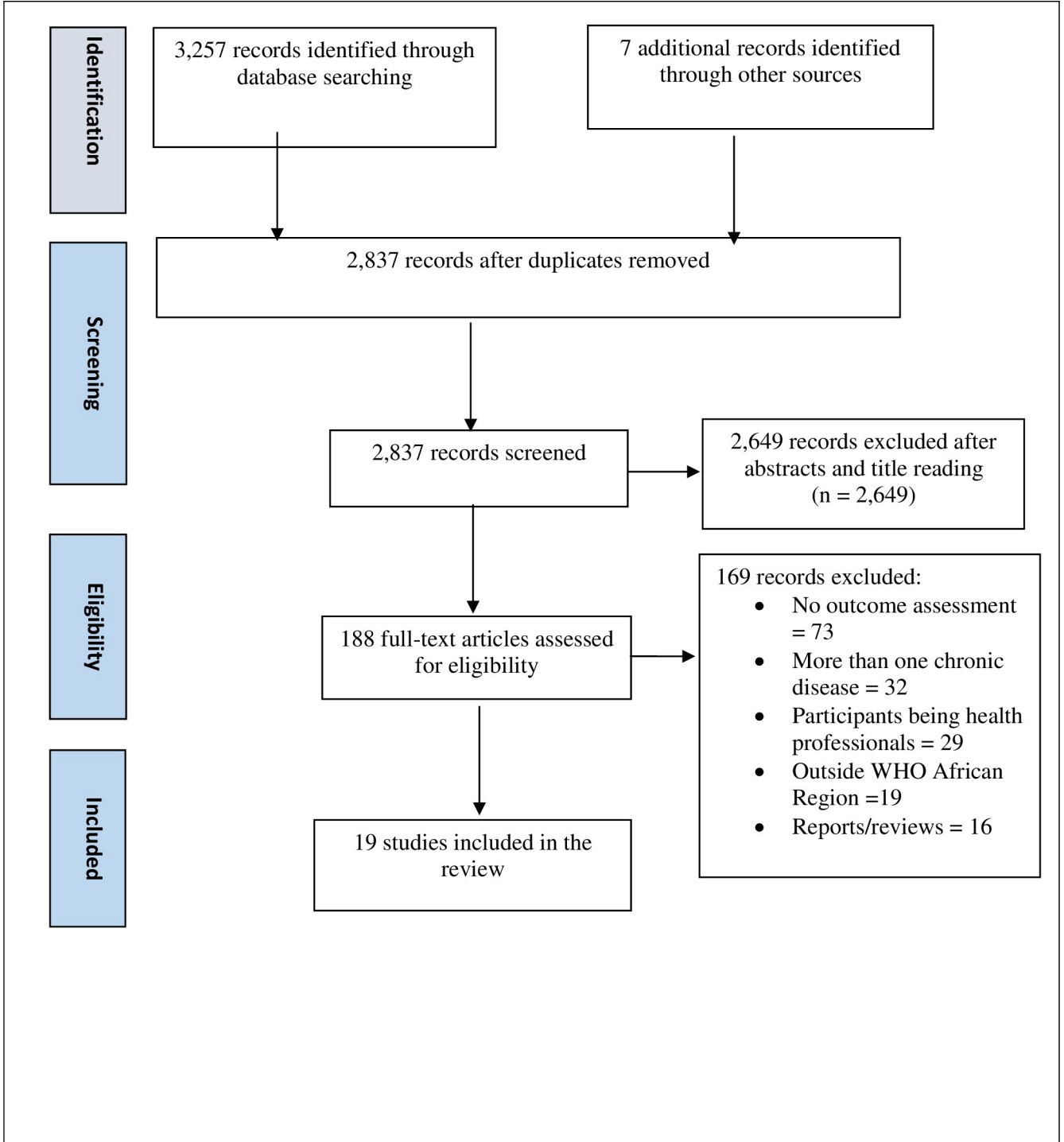

**Fig 1. Literature search flow diagram.**

**Table 1. Quality assessment of the included studies.**

| Study | Overall Rating |
|---|---|
| Assah et al. [25] | Moderate |
| Bett [26] | Weak |
| Debussche et al. [27] | Strong |
| Gill et al. [28] | Moderate |
| Hailu et al. [29] | Moderate |
| Hailu et al. [30] | Moderate |
| Mash et al. [31] | Moderate |
| Muchiri et al. [32] | Moderate |
| Afemikhe & Chipps [33] | Weak |
| Essien et al. [34] | Moderate |
| Park et al. [35] | Weak |
| Asante et al. [36] | Moderate |
| Price et al. [37] | Weak |
| Amendezo et al. [38] | Moderate |
| Muchiri et al. [39] | Moderate |
| MakkiAwouda et al. [40] | Weak |
| Baumann et al. [41] | Moderate |
| van der Does & Mash [42] | Weak |
| Gathu et al. [43] | Weak |

## Study quality

The overall average (SD) quality score of the 19 studies was 1.74 (±0.41) ranging from 1.17 to 2.67. Eleven studies were rated as having moderate methodological quality, seven as weak and only one as strong (Table 1). Scores for individual studies on the EPHPP six domains of study quality are shown in S4 Appendix.

## Characteristics of the identified studies

Details of the included studies are summarized in Table 2. The majority of the studies were conducted in South Africa (n = 6), followed by Kenya (n = 3), Ethiopia and Nigeria (n = 2 each). The rest were conducted in Cameroon (n = 1), Mali (n = 1), Ghana (n = 1), Rwanda (n = 1), Sudan (n = 1) and Uganda (n = 1). Eleven studies were randomized controlled trials (RCTs) or studies with random assignment, six were quasi-experimental designs, one was mixed methods, and one was observational cohort study. The studies were conducted between 2008 and 2020. Sixteen studies included patients with type 2 diabetes, while two included participants with both type 1 and type 2 diabetes. One study did not specify the type of diabetes patients who participated in the DSME intervention. The total sample size for the 19 studies was 3,959, with a mean age of 54.2 years (excluding 6 studies which did not provide information on the mean age of the study participants).

## Intervention characteristics

The interventions differed in their design, including strategies used, delivery mode and duration (Table 3). The DSME interventions could be described as individually oriented, group-based, individually oriented & group-based, and information technology-based education programs. Fourteen interventions [25–27, 29–32, 34, 35, 37–39, 41, 42] were group-based, utilizing group meetings, illustrative pictures and handbooks, as well as experience-sharing and

**Table 2. Details of the reviewed papers.**

| Author, Year | Country | Study Design and Purpose | Sample Description |
|---|---|---|---|
| Assah et al. [25] | Cameroon | RCT: | 192 subjects with poorly controlled type 2 diabetes (intervention = 96, control = 96); 45 men and 51 women for both groups; no age difference between intervention and control groups (57.1 vs. 57.2 years) |
| | | To examine the effectiveness of a community-based multilevel peer support intervention on improving HbA1c, blood pressure and lipids in patients with T2DM | |
| Bett [26] | Kenya | Non-randomized experimental design: | 123 adults with T2DM (intervention = 63, control = 60), more females in control (56.7%) than in intervention (47.6%) |
| | | To determine if a structured diabetes education intervention for T2DM patients would increase their diabetic knowledge, self-efficacy, and reduce their HbA1c levels | |
| Debussche et al. [27] | Mali | RCT: | 151 adults with T2DM (intervention = 76, control = 75), 76% women, mean age 52.5years |
| | | To evaluate the effectiveness of peer-led self-management education in improving glycemic control in T2DM patients | |
| Gill et al. [28] | South Africa | A pre-post design: | 284 type 1 & 2 diabetes patients (96% type 2), mean age = 56 years, 80% female |
| | | To set up and evaluate a nurse-led protocol and education-based system | |
| Hailu et al. [29] | Ethiopia | Before-and-after controlled study design, with random assignment: | 220 type 2 DM patients (intervention = 116, control = 104), mean age = 54.5 years |
| | | To determine the effects of DSME on clinical outcomes among T2DM patients | |
| Hailu et al. [30] | Ethiopia | RCT: | 220 T2DM patients (intervention = 116, control = 104) |
| | | To develop and test the effectiveness of a multifaceted, nurse-led DSME program | |
| Mash et al. [31] | South Africa | A clustered RCT: | 1,570 patients with T2DM (intervention = 710, control = 860), 73.8% male, mean age = 56.1 years |
| | | To evaluate the effectiveness of group education for people with T2DM | |
| Muchiri et al. [32] | South Africa | RCT: | 82 adults (aged 40–70 years) with T2DM (intervention = 41, control = 41), mean age = 58.8 years |
| | | To evaluate the effect of a participant-customized nutrition education program on HbA1c, blood lipids, blood pressure, BMI and dietary behaviors in patients with T2DM | |
| Chipps and Afemikhe [33] | Nigeria | A quasi-experimental study: | 28 adults with T2DM (intervention = 15, control = 13), 11 males and 17 females, mean age = 56.7 years |
| | | To pretest whether a structured multidisciplinary patient centered DSME program for type 2 diabetes would improve selected primary and secondary diabetes outcome measures | |
| Essien et al. [34] | Nigeria | Un-blinded, parallel-group, individually-RCT: | 118 type 1 and type 2 diabetes patients (intervention = 59, control = 59), Male = 47, female = 71, mean age = 52.7 years |
| | | To evaluate whether an intensive and systematic DSME programme, using structured guidelines, improved glycemic control | |
| Park et al. [35] | Kenya | A pre-post implementation study: | 148 adults aged ≥ 18 years and diagnosed with type 1 or type 2 DM |
| | | To evaluate the impact of a 6-month diabetes self-management support (DSMS) intervention on diabetes mellitus | |
| Asante et al. [36] | Ghana | A pilot RCT: | 60 adults aged ≥ 18 years with T2DM (intervention = 30, control = 30), 78.33% female (n = 47) |
| | | To compare diabetes care as usual to a mobile phone call intervention | |
| Price et al. [37] | South Africa | Single-center, observational cohort study: | 80 patients with T2DM, mean ±SD age 56 ±11 years, 70% female |
| | | To determine the long-term glycemic outcome of a structured nurse-led intervention program for T2DM patients | |
| Amendezo et al. [38] | Rwanda | An un-blinded, parallel-group, RCT: | 223 adults aged ≥ 21 years with T2DM (intervention = 115, control = 108), mean age 51.5 (+/-11) years, 71% female |
| | | To assess the efficacy of a structured lifestyle education program | |
| Muchiri et al. [39] | South Africa | RCT: | 82 adults, aged 40–70 years, with poorly controlled T2DM (intervention = 41, control = 41), mean age = 58.8 years (SD 7.7 years) |
| | | To evaluate the effect of a nutrition education program on diabetes knowledge | |

(*Continued*)

**Table 2.** (Continued)

| Author, Year | Country | Study Design and Purpose | Sample Description |
|---|---|---|---|
| MakkiAwouda et al. [40] | Sudan | Quasi-experimental study design: | 152 patients with diabetes (58 male, 94 female) |
| | | To determine the effects of health education on the control and improvement in the health status of diabetes patients | |
| Baumann et al. [41] | Uganda | A pre-post quasi-experimental study: To test the feasibility of a peer intervention to improve self-care behaviors and health status of diabetes patients | 46 adults aged $\geq$ 18 years with T2DM |
| van der Does & Mash [42] | South Africa | A mixed methods study: To evaluate a group education program for patients with T2DM | 84 patients with T2DM (81% female), mean age = 51.6 years (SD 9.2) |
| Gathu et al. [43] | Kenya | Non-blinded RCT: | 96 T2DM patients (intervention = 55, control = 41), mean age = 48.8 (SD 9.8) years |
| | | To assess the effects of DSME in comparison to usual diabetes care | |

RCT Randomized Controlled Trial, DM Diabetes Mellitus, DSME Diabetes Self-Management Education, T2DM Type 2 Diabetes Mellitus

take-home activities. Two interventions [28, 33] combined both group and individually-oriented mode of delivery. For instance, in the intervention evaluated by Gill et al. [28], a full program of group education was delivered in the first three monthly sessions, after which selected topics were reinforced at individual clinic visits. Two interventions [40, 43] were individually oriented, employing a one-to-one mode of delivery. Only one information technology-based intervention [35] was reviewed. This intervention was delivered through 16 mobile phone calls, with a mean call duration of 12 minutes.

Length of the interventions varied, ranging from 4 weeks [42] to 48 months [37]. The majority of the programs (63.2%) were delivered at tertiary care facilities [26, 29, 30, 34, 36, 38] and clinics [28, 35, 37, 41–43]. Five interventions [27, 31, 32, 39, 40] were delivered at health centers, and one at both tertiary and secondary care facilities [33]. One intervention [25] did not have a specific place of delivery. It is important to note that within health service delivery in most African countries, clinics and health centers both focus on primary health or outpatient care. However, in the WHO African Region, health centers are community-based health facilities focusing on general primary care, while most clinics are specialized health facilities focusing on specific diseases and conditions.

A range of health and non-health professionals delivered the DSME interventions. Six interventions [26, 33, 34, 38, 41, 42] were delivered by an interdisciplinary team of health professionals, such as doctors/physicians, nurses, dietitians/nutritionists, and medical social workers. Five interventions [28–30, 36, 37] were delivered by nurses, three [25, 27, 35] by peer educators, two [40, 43] by diabetes health educators, two [32, 39] by dietitians and one [31] by health promoters—paid non-medically trained professionals whose work is to promote public health.

Regarding theoretical underpinning, seven of the 19 studies indicated specific behavior theories guiding their interventions. These included: the Health Belief Model [26, 32], Socio-Constructivist Theory [27], Bandura's Social Cognitive Theory of Behavior [28, 32, 33], Motivational Interviewing [31, 32], the Knowledge Attitude Behavior Model [32], and Self-Determination Theory [33].

## Study outcomes

Outcomes of the reviewed studies are summarized in Table 4.

**Table 3. Characteristics of the DSME interventions.**

| Author, Year | Intervention | Setting | Provider of Education | Theoretical Underpinning | Program Length |
|---|---|---|---|---|---|
| Assah et al. [25] | A peer support intervention implemented through group meetings, personal encounters between peer supporters and group members and telephone calls | Locations related to each group's common affinity | Peer Educators | | 6 months |
| Bett [26] | A structured education once every week for three weeks and three months follow-ups | Hospital | Nurses, Dieticians and Doctors | The Health Belief Model (HBM) | 4.5 months |
| Debussche et al. [27] | A 1-year culturally tailored structured patient education (3 courses of 4 sessions) | Community Health Center | Trained Peer Educators | The 'Learning Nests' approach, derived from Socio-Constructivist Theory | 12 months |
| | Themes addressed were cardiovascular risk management, food intake, exercise, and blood glucose and insulin management | | | | |
| Gill et al. [28] | A treatment algorithm and education system developed into primary health clinics | Primary Care Clinic | Nurses | Bandura's Social Cognitive Theory of Behavior | 18 months |
| Hailu et al. [29] | Six educational sessions supported with illustrative pictures, handbooks and fliers customized to local conditions | University Medical Centre | Nurses | | 9 months |
| Hailu et al. [30] | Six interactive diabetes SME sessions supported by an illustrative handbook and fliers, experience-sharing, and take-home activities | University Medical Centre | Nurses | | 9 months |
| Mash et al. [31] | Four 60-minute sessions of group education focusing on understanding diabetes, living a healthy lifestyle, understanding the medication, and avoiding complications | Community Health Center | Health Promoters | Motivational Interviewing | 4 months |
| Muchiri et al. [32] | Eight weekly (2–2·5 hours) group nutrition education and follow-up sessions | Community Health Center | Dietitians | The Social Cognitive Theory, the Health Belief Model and the Knowledge Attitude Behavior Model | 12 months |
| Afemikhe & Chipps [33] | A five-week multidisciplinary education program utilizing group discussions, individual counselling, multimedia teaching, motivational interviewing, telephone calls by nurses and goal-setting charts for feedback | Hospital (one tertiary & one secondary) | Nurses, Dietitians and Medical Social Workers | Self-Determination Theory, Social Cognitive Theory and the Motivational Interviewing Framework | 5 weeks |
| Essien et al. [34] | Twelve structured teaching sessions lasting around two hours each, attended fortnightly over a six-month period. | Tertiary Hospital | Doctors and Nurses | | 6 months |
| Park et al. [35] | A 6-month peer-led bimonthly group educational program on self-empowerment and problem-solving surrounding behavioral modification and self-management skills | Peri-Urban and Rural Diabetes Mellitus Clinics | Peer Educators | | 6 months |
| Asante et al. [36] | A 12-week mobile phone call intervention (2 calls per week for the first 4 weeks, followed by a weekly call for the following 8 weeks, totaling 16 calls) | Tertiary Hospital | Nurses | | 12 weeks |
| Price et al. [37] | A structured empowerment-based diabetes education delivered in groups and regularly reinforced | Primary Health Clinics | Nurses | | 48 months |
| Amendezo et al. [38] | Group education sessions focusing on: setting balanced diabetic diet, regular physical activity, cessation of smoking and alcohol abuse, adherence to medications, diabetic complications screening and treatment, self-management of hypoglycemia and hyperglycemia, and stress management | Tertiary Hospital | Physicians, Nurses, Nutritionists | | 12 months |
| Muchiri et al. [39] | Eight-weekly group education (2 to 2.5 hours each) with follow-up sessions (4 monthly meetings and 2 bi-monthly meetings each lasting 1.5 hours), and vegetable gardening (demonstration of sowing/transplantation of vegetables) | Community Health Center | Dietitians | Knowledge Attitude Behaviour (KAB) model and the Health Belief Model (HBM) | 12 months |

(*Continued*)

**Table 3.** (Continued)

| Author, Year | Intervention | Setting | Provider of Education | Theoretical Underpinning | Program Length |
|---|---|---|---|---|---|
| MakkiAwouda et al. [40] | A one–to—one educational intervention focusing on patho-physiological view, modalities of treatment, and identifications, prevention and treatment of acute complications | Health Center | Diabetes Health Educators | | 3 months |
| Baumann et al. [41] | A 4-month peer support intervention in which participants were trained in diabetes self-care | Diabetes Clinic | Physicians and Nurses | | 4 months |
| van der Does & Mash [42] | Four sessions of an hour each of group education; topics addressed: knowledge about diabetes, complications and treatment, healthy lifestyle and how to apply diabetes knowledge in day-to-day life | Primary Care Clinic | Dietitian, Health Promoter and Physician | | 4 weeks |
| Gathu et al. [43] | An individualized structured DSME intervention using an empowerment and interactive teaching model, with a focus on behavioral assessment, goal-setting and problem-solving | Primary Care Clinic | Certified Diabetes Educators | | 6 months |

**Learning outcomes.** Two indicators of learning outcome were assessed: self-efficacy (confidence in self-management) and diabetes knowledge. Self-efficacy was assessed in four studies. Bett [26] reported significant improvement in self-efficacy among the intervention group compared with the control ($F_{(1, 117)} = 14.342$, $p<0.001$). However, no significant improvements were observed in the other three studies [30, 31, 41].

Six studies measured the effect of DSME interventions on diabetes knowledge. Five studies [26, 27, 30, 39, 40] demonstrated improvements. For instance, Muchiri et al. [39] reported that the intervention group had higher mean diabetes knowledge scores of +0.95 ($p = 0.033$) and + 2.05 ($p < 0.001$) at 6 and 12 months respectively. MakkiAwouda et al. [40] also indicated that the average knowledge for the nature of diabetes significantly improved from 0.9408 to 1.74 (t-value = 7.38, p = 0.000). One study [35] reported no improvement in diabetes knowledge.

**Behavioral outcomes.** Outcomes for diabetes-related behaviors reported in the included studies were: dietary practices, physical activity/exercise, foot care, blood glucose self-monitoring, smoking, alcohol consumption, and medication adherence. Dietary practices were reported in seven studies; four [23, 30, 39, 40] demonstrated significant improvements, three [27, 31, 36] reported no significant changes. Physical activity or exercise was measured in six studies [25, 30, 31, 36, 41, 42]; only two [25, 42] demonstrated significant positive effects. Foot care practices were assessed in four studies; three [25, 36, 42] reported significant improvements, one [31] showed no significant positive change. Self-monitoring of blood glucose was an outcome measure in two studies [30, 36]; all demonstrating no significant improvements. Tobacco smoking was measured in three studies; one [42] reported positive effect, two [30, 41] demonstrated no significant positive effects. Alcohol consumption was measured in one study [30], but the authors reported no statistically significant positive effect. Finally, medication adherence was assessed in three studies [31, 36, 42]; all reporting no significant positive changes.

**Clinical outcomes.** Clinical outcome indicators assessed included: glycated hemoglobin (HbA1c), body mass index (BMI)/weight, blood pressure, lipid profiles, waist circumference, and blood sugar/glucose. HbA1c, the most common outcome, was measured in 14 studies. Of these, ten [25–28, 34–38, 41] reported significant improvements in patients' HbA1c levels. For instance, Debussche et al. [27] found a decrease in HbA1c levels of 1.05% in the intervention group compared with 0.15% in the control group (p = 0.006). Gill et al. [28] also reported that HbA1c improved from 11.6 ± 4.5% at baseline to 8.7 ± 2.3% at 3 months and 7.7 ± 2.0% at 18

**Table 4. Outcomes of the DSME interventions.**

| Author, Year | Outcome Measures | | | Results |
|---|---|---|---|---|
| | Learning Outcomes | Behavioral Outcomes | Clinical Outcomes | |
| Assah et al. [25] | - | Diet, exercise, foot care | HbA1c, BMI, FBS, cholesterol, blood pressure, HDL | • Significant reduction in HbA1c in the intervention group [−33 mmol/mol (−3.0%)] compared with controls [−14 mmol/mol (−1.3%)], P < 0.001 |
| | | | | • Significant reductions in FBS (−0.83 g/l P < 0.001), cholesterol (−0.54 g/l P < 0.001), HDL (−0.09 g/l, P < 0.001), BMI (−2.71 kg/m$^2$ P < 0.001) and diastolic pressure (−6.77 mmHg, P < 0.001) |
| | | | | • Diabetes self-care behaviors (diet, exercise and foot care) in the intervention group also improved significantly |
| Bett, [26] | Self-efficacy, diabetes knowledge | - | HbA1c | • The experimental group had significant reduction levels of HbA1c (F$_{(1, 122)}$ = 9.989, p = 0.002), and improved diabetes knowledge (t = 7.218, p = <0.001) and self-efficacy (F$_{(1, 117)}$ = 14.342, p<0.001) |
| Debussche et al. [27] | Knowledge score | Dietary practices | HbA1c, weight, BMI, waist circumference, SBP & DBP | • A decrease in HbA1c levels of 1.05% (SD = 2.0; CI95%: 1.54; -0.56) in the intervention group compared with 0.15% (SD = 1.7; CI95%: -0.56; 0.26) in the control group, p = 0.006 |
| | | | | • Mean BMI change was -1.65 kg/m2 (SD = 2.5; CI95%: -2.25; -1.06) in the intervention group and +0.05 kg/m2 (SD = 3.2; CI95%: -0.71; 0.81) in the control group, p = 0.0005 |
| | | | | • Mean waist circumference decreased by 3.34 cm (SD = 9.3; CI95%: -5.56; -1.13) in the intervention and increased by 2.65 cm (SD = 10.3; CI95%: 0.20; 5.09) in the control group, p = 0.0003 |
| | | | | • SBP and DBP improved in the intervention group than in the control group. Patients' knowledge scores improved |
| | | | | • No positive change in the diet diversity score as a crude index of diet quality was recorded, but qualitative changes in the diet were noted |
| Gill et al. [28] | - | - | HbA1c, BMI, hypoglycemia | • HbA1c improved from 11.6 ± 4.5% at baseline to 8.7 ± 2.3% at 3 months and 7.7 ± 2.0% at 18 months |
| | | | | • Significant increase in BMI |
| | | | | • No significant change in hypoglycemia |
| Hailu et al. [29] | - | - | HbA1c, FBS, SBP, DBP | • Mean HbA1c significantly reduced by 2.88% within the intervention group and by 2.57% within the control group, but between group differences were not statistically significant |
| | | | | • Adjusted end-line FBS, SBP, and DBP were significantly lower in the intervention group, by 27 ± 9 mg/dL, 12 ± 3, and 8 ± 2 mmHg respectively |
| Hailu et al. [30] | Diabetes knowledge, self-efficacy | Self-care behaviors | - | • Significant mean difference in diabetes knowledge (p = 0.044), dietary recommendations (p = 0.019) and foot care performed (p = 0.009) in the intervention group |
| | | | | • No significant differences within or between groups in the other self-care behaviors (exercise, glucose self-monitoring, smoking, alcohol consumption) or in diabetes self-efficacy |
| Mash et al. [31] | Self-efficacy | Physical activity, use of diet plan, use of medication, foot care, & smoking | HbA1c, weight, waist circumference, SBP & DBP | • No significant improvement in the outcomes, apart from a significant reduction in mean SBP (-4.65 mmHg, 95% CI 9.18 to -0.12; P = 0.04) and DBP (-3.30 mmHg, 95% CI -5.35 to -1.26; P = 0.002) |

*(Continued)*

**Table 4.** (*Continued*)

| Author, Year | Outcome Measures | | | Results |
|---|---|---|---|---|
| | **Learning Outcomes** | **Behavioral Outcomes** | **Clinical Outcomes** | |
| Muchiri et al. [32] | - | Dietary behaviors | HbA1c, blood lipids, blood pressure, BMI | • No significant group difference in HbA1c (−0·64%, P = 0·15 at 6 months and −0·63%, P = 0·16 at 12 months) |
| | | | | • No significant group differences in BMI, lipid profile, and blood pressure |
| | | | | • Starchy-food intake was significantly lower in the intervention group, 9·3 v. 10·8 servings/d (P = 0·005) at 6 months and 9·9 v. 11·9 servings/d (P = 0·017) at 12 months |
| Afemikhe & Chipps [33] | - | - | FBS, BMI, SBP | • The intervention group had significantly lower FBS (p = 0.01) and BMI scores (.025) than the control group, but only FBS differed significantly between the two groups (p = .012) |
| | | | | • No significant group difference in SBP (p = .467) |
| Essien et al. [34] | - | - | HbA1c | • Participants in the intervention group had significantly lower HbA1c levels compared to participants in the control group, with a mean estimated HbA1c difference of -1.8 (95% CI: -2.4 to -1.2) |
| Park et al. [35] | Diabetes knowledge | - | HbA1c, SBP, BMI | • Improvement in HbA1c ($\beta$ -0.17, SE 0.09, P = 0.05) and SBP ($\beta$ -5.67, SE 1.64, P = 0.001, with a median decrease from 132.4 mmHg to 127.5 mmHg) |
| | | | | • No changes in diabetes knowledge and BMI |
| Asante et al. [36] | - | Diet, exercise, medication taking, foot care, and blood glucose monitoring | HbA1c | • HbA1c was significantly lower in the intervention group compared to the control group. The difference in mean HbA1c in the control group rose by +0.26 ± 1.30% (*P* = .282; 95% CI, −0.23 to 0.75), whereas that of the intervention group reduced by −1.51 ± 2.67% (*P* = .004; 95% CI, −2.51 to −0.51) |
| | | | | • Foot care practices improved |
| | | | | • No significant improvements in the other outcomes |
| Price et al., [37] | - | - | HbA1c and BMI | • HbA1c fell significantly to 8.1 ± 2.2% at 6 months and 7.5 ±2.0% at 18 months. At 24 months, it had risen to 8.4 ± 2.3%, and at 4 years post-intervention it was 9.7± 4.0% (still significantly lower than baseline, P = 0.015) |
| | | | | • BMI at 6 and 18 months was significantly higher than at baseline (both P < 0.01), but the 48-month value was not significantly different from 0 months |
| Amendezo et al. [38] | - | - | HbA1c, SBP, DBP, BMI, FBG | • Statistically significant between group difference in change in HbA1c (p <0.001), FBG (p <0.001), SBP (p <0.005), DBP (p <0.02) and BMI (p <0.001) |
| Muchiri et al. [39] | Diabetes knowledge | - | - | • The intervention group had higher mean diabetes knowledge scores + 0.95 (*p* = 0.033) and + 2.05 (*p* < 0.001) at 6 and 12 months respectively |
| MakkiAwouda et al. [40] | Diabetes knowledge | - | - | • The average knowledge for the nature of diabetes significantly improved from 0.9408 to 1.74 (t-value = 7.38, p = 0.000) |
| Baumann et al. [41] | Confidence in self-management | Diet (healthy eating), physical activity | HbA1c, SBP, DBP, BMI | • The average DBP dropped from 85.39 to 76.27 mmHg (p<0.001), and the average HbA1c values changed from 11.10 to 8.31% (p<0.005) |
| | | | | • Average BMI values did not change |
| | | | | • Of the health behaviors measured, only healthy eating significantly changed in a positive direction from pre-intervention to post-intervention, p<0.005. Confidence in self-management did not change |

(*Continued*)

**Table 4.** (Continued)

| Author, Year | Outcome Measures | | | Results |
|---|---|---|---|---|
| | **Learning Outcomes** | **Behavioral Outcomes** | **Clinical Outcomes** | |
| van der Does & Mash [42] | - | Diet, physical activity, foot care, medication adherence | - | • Significant improvement in adherence to diet, physical activity, foot care |
| | | | | • No self-reported change in adherence to medication |
| | | | | • Tobacco smoking reduced from 25% (21/84) to 18% (15/84) ($p = 0.08$) |
| Gathu et al. [43] | - | - | HbA1c, BMI blood pressure | • No significant difference was noted in HbA1c between the two groups, with a mean difference of 0.37 (95% confidence interval: -0.45 to 1.19; $p = 0.37$) |
| | | | | • Blood pressure and BMI did not change from baseline to 6 months follow-up |

HbA1c Glycated Hemoglobin, BMI Body Mass Index, FBS Fasting Blood Sugar, FBG Fasting Blood Glucose, HDL High Density Lipoprotein, SBP Systolic Blood Pressure, DBP Diastolic Blood Pressure

months. Four studies [29, 31, 32, 43] demonstrated no significant reduction in HbA1c. Hailu et al. [29] indicated that although mean HbA1c significantly reduced by 2.88% within the intervention group and by 2.57% within the control group, between group difference was not statistically significant. Muchiri et al. [32] also reported no significant group difference in HbA1c (−0·64%, P = 0·15 at 6 months and −0·63%, P = 0·16 at 12 months).

With the other clinical outcomes, changes in weight or BMI were measured in 11 studies; three [25, 27, 38] reported statistically significant positive changes, eight [28, 31–33, 35, 37, 41, 43] showed no significant effects. Blood pressure was assessed in ten studies; seven [25, 27, 29, 31, 35, 38, 41] reported statistically significant positive effects, three [32, 33, 43] demonstrated no significant reduction in blood pressure levels. Lipid profiles, including cholesterol and high-density lipoprotein, were the outcome measures of two studies; one [25] reported statistically significant positive effect, one [32] showed no significant, positive effect. Two studies reported on waist circumference; one [27] had positive effect, the other [31] showed no significant effect. Blood glucose or sugar was an outcome measure of five studies. Four [25, 29, 33, 38] of these studies reported statistically significant positive effects, while one [28] indicated no significant effect.

## Discussion

Although self-management education has become an integral and a vital component of diabetes care, its implementation in Africa has not been well documented [19, 20]. This scoping review was conducted to provide the state of the science of DSME interventions in the WHO African Region and to assess program outcomes. The interventions identified were individually oriented, group-based, individually oriented & group-based, and information technology-based DSME programs. Outcomes of the interventions were mixed. While the majority yielded significant positive results on HbA1c, diabetes knowledge, blood pressure, blood sugar and foot care practices; few demonstrated positive outcomes on self-efficacy, BMI, physical activity; self-monitoring of blood glucose, medication adherence, smoking and alcohol consumption. Also, the majority of the interventions were more effective on the learning and clinical outcomes compared with the behavioral outcomes.

While 14 studies reported positive results on HbA1c outcome suggesting the need for a meta-analysis, the studies were not homogeneous enough to conduct a meta-analysis. For,

instance, in terms of study design, 11 out of the 19 studies were randomized controlled trials (RCTs) although 14 studies assessed HbA1c. Also, the 11 studies that measured the same outcome (HbA1c) were so diverse in terms of the subjects involved and the interventions implemented (see Tables 2 and 3 for descriptions of the subjects and the DSME interventions). Thus, combining the studies that differ substantially in a meta-analysis could yield a meaningless summary result. Above all, the aim of this study was to describe the nature and types of DSME interventions that have been implemented in the WHO African Region.

In spite of the mixed outcomes reported by the included studies, the findings support studies conducted in the US [44], Europe [45] and other Western countries [46] that DSME interventions are effective in improving patients' HbA1c levels. Ten [25–28, 34–38, 41] of the 14 interventions that assessed patients' HbA1c levels reported statistically significant decreases. The remaining four [29, 31, 32, 43] also reported decreases in HbA1c levels, except that the differences between the intervention and the control groups were not statistically significant.

It has been asserted that if educational sessions are reinforced periodically, benefits could be sustained for a longer period [47, 48]. This is supported by one of the reviewed interventions, which involved a structured empowerment-based education delivered in groups and regularly reinforced [37]. The authors reported that at 4 years post-intervention, HbA1c levels were still significantly lower than at baseline ($p = 0.015$).

We observed that the majority of the educational interventions were delivered in group settings. This is consistent with the literature that group-based education has become the preferred format for delivering self-management education and medical nutrition therapy interventions [49]. Group-based education has been found to be more cost-effective and efficient compared to individualized, educational interventions [47, 50–52]. It has however been argued that since people with diabetes have different learning needs, it is essential for patients to be offered the option of whether they prefer learning in a group or individually so as to cater for these varied needs [53]. For instance, the National Institute for Health and Care Excellence (NICE) in UK suggests that although people with diabetes should be offered group education as the preferred option, alternative individual education should be provided for those who are unable or unwilling to attend group education sessions [54].

It is well documented that interventions designed to influence health behavior (e.g., diabetes self-management) are more likely to be beneficial when they are grounded in theories [55]. However, only a few interventions (7 out of 19) included in this review were guided by behavioral change theories. Although we did not observe any marked difference (in terms of positive results on the participants) between the interventions guided by theories and those not guided by any specific behavior theory, grounding an intervention in a theory helps in identifying targets for change, as well as informing evaluation and providing a roadmap for future refinement and dissemination [55].

One key outcome measure that was not assessed in any of the included studies is diabetes-related healthcare utilization measured in terms of hospital admissions, length of stay, emergency department admissions, visits to specialist clinics, and others [56]. Self-management interventions have gained prominence because of their potential to contribute significantly to efficient healthcare delivery by increasing patient engagement in care, improving the uptake of preventive practices and reducing reliance on formal healthcare services [56]. Thus, the success of a DSME intervention is also measured by its ability to reduce healthcare use.

The findings presented in this review should be interpreted in light of the weak to moderate quality of evidence examined. Potential biases in the methodological conduct of the studies included: sample not representative of target population [33, 37, 42], marked differences in characteristics between intervention and control groups [26, 35, 40, 43], lack of participant blinding [26, 33, 35, 37, 40, 42, 43], and issues with withdrawals and drop-outs (attrition bias)

[33, 35, 37, 38, 43]. Also, because we did not restrict the review to only studies published in peer-reviewed academic journals, some of the included papers were either not peer-reviewed [26] or lacked a rigorous peer review process [33, 40].

## Study limitations

The limitations of this review are worth acknowledging. First, we included only articles published in English journals from 2000 to 2020, thus excluding useful information that may be in other languages or may have been published before 2000. Also, the included studies had different research designs, such as randomized controlled trial, quasi-experimental study, mixed methods and observational cohort study. This could have implications for interpretation of the findings synthesized from the studies. Again, the inclusion of only 19 studies in the review is an indication that the conclusions drawn are based on limited data. Furthermore, the review did not include qualitative evaluation of DSME interventions. Despite these limitations, we believe the review provides useful information that may inform the development and implementation of DMSE interventions in Africa and other developing countries.

## Conclusion and future directions

The limited studies available indicate that DSME interventions in the WHO African Region have mixed effects on patient behaviors and health outcomes. That notwithstanding, the majority of the interventions demonstrated statistically significant positive effects on HbA1c, the main outcome measure in most DSME intervention studies.

This review is important as it has made known gaps that need to be addressed for effective development and implementation of DSME interventions in Africa, particularly countries in the WHO African Region. First, few studies on DSME have been conducted in the WHO African Region. There is therefore the need to scale up both observational and interventional studies on DSME in the Region. Second, self-management education is about behavior change, thus the development and implementation of interventions should be guided by behavior change theories. Third, one of the goals of a self-management intervention is to reduce healthcare cost through a reduction in healthcare use. Thus, future DSME interventions in the WHO African Region should consider assessing this key outcome measure. Finally, there is the need to improve the methodological rigor of future DSME studies in the Region. Overall, we judged the quality of the included studies to be moderate (1.74, rang of 0–3), with about 37% of them being rated as weak.

As qualitative studies were not included in this study, we recommend that future research should focus on qualitative evaluation of DSME in the WHO African Region.

## Supporting information

**S1 Appendix. PRISMA extension for Scoping Reviews (PRISMA-ScR) checklist.**
(DOCX)

**S2 Appendix. Sample search strategy.**
(DOCX)

**S3 Appendix. Quality assessment tool used to evaluate the methodological quality of the included studies.**
(DOCX)

**S4 Appendix. Quality assessment of the included studies.**
(DOCX)

## Acknowledgments

Rev Dr. Samuel Kofi Agyei of Presbyterian University College of Ghana for critically appraising this paper. We also thank the staff of the Planning Unit, Komfo Anokye Teaching Hospital, for the secretariat services provided.

## Author Contributions

**Conceptualization:** Emmanuel Kumah.

**Data curation:** Emmanuel Kumah, Godfred Otchere, Samuel Egyakwa Ankomah, Kofi Aduo-Adjei.

**Formal analysis:** Emmanuel Kumah, Godfred Otchere, Samuel Egyakwa Ankomah, Kofi Aduo-Adjei.

**Methodology:** Emmanuel Kumah, Godfred Otchere, Samuel Egyakwa Ankomah, Adam Fusheini, Collins Kokuro, Kofi Aduo-Adjei.

**Supervision:** Emmanuel Kumah.

**Writing – original draft:** Emmanuel Kumah.

**Writing – review & editing:** Emmanuel Kumah, Adam Fusheini, Collins Kokuro, Joseph A. Amankwah.

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
