## [Decision Letter · Decision Letter 0]

16 Jun 2021

PONE-D-21-04506

Diabetes self-management education interventions in the WHO African Region: a scoping review

PLOS ONE

Dear Dr. Kumah,

Thank you for submitting your manuscript to PLOS ONE. After careful consideration, we feel that it has merit but does not fully meet PLOS ONE’s publication criteria as it currently stands. Therefore, we invite you to submit a revised version of the manuscript that addresses the points raised during the review process.

We look forward to receiving your revised manuscript.

Kind regards,

Lee-Ling Lim

Academic Editor

PLOS ONE

Additional Editor Comments:

1) Line 164-165: How did the authors exclude articles on diabetes and comorbidity? It’s very common for people with diabetes to have one or more comorbidity e.g., hypertension, cardiovascular disease. Would exclusion of these articles affect the results?

2) Line 170: What was the rationale of excluding articles with 2 or more DSME interventions? Given that the authors aimed to determine the types and effects of DSME interventions, would including these articles provide more robust results?

3) Has this systematic review been registered in PROSPERO?

4) Line 315: Since there were 14 studies with HbA1c outcome, a meta-analysis should be performed, taking into account the study designs.

5) Line 325-327: Similar comments applied to the changes in weight, BMI and blood pressure.

6) Figure 1: Please provide the reasons of excluding 169 articles.

7) How many articles were solely targeting either type 1 diabetes or type 2 diabetes? Could the outcome assessment be stratified by the type of diabetes?

Journal Requirements:

2. We note that this manuscript is a systematic review or meta-analysis; our author guidelines therefore require that you use PRISMA guidance to help improve reporting quality of this type of study. Please upload copies of the completed PRISMA checklist as Supporting Information with a file name “PRISMA checklist”.

Reviewers' comments:

Reviewer's Responses to Questions

**Comments to the Author**

1. Is the manuscript technically sound, and do the data support the conclusions?

Reviewer #1: Yes

2. Has the statistical analysis been performed appropriately and rigorously? 

Reviewer #1: Yes

3. Have the authors made all data underlying the findings in their manuscript fully available?

Reviewer #1: Yes

4. Is the manuscript presented in an intelligible fashion and written in standard English?

Reviewer #1: Yes

5. Review Comments to the Author

Reviewer #1: Minor comments:

Reference #2 is not appropriate for line 68 as this article is referencing another source. Please reference appropriately.

Reference #5 is not appropriate for line 73, line 82 & line 86 as this article is referencing other sources. Please reference appropriately.

Line 128: is that SI or S1?

Line 170: Could the authors explain the rationale for this exclusion point?

Line 171: could have included qualitative studies or mixed methods – if the outcome is behavioural change – and this would have more depth to the results while still keeping to the study objectives. I also noted you included one 1 mixed methods study (line 240).

Can the authors clarify the difference between ‘clinics’ and ‘health centers’ Line 264-265? In some countries, these are the same (only difference in name)

Please also explain what is a ‘health promoter’ Line 273 (paid or non-paid non-medically trained individuals? Volunteers?)

6. PLOS authors have the option to publish the peer review history of their article (what does this mean?). If published, this will include your full peer review and any attached files.

Reviewer #1: **Yes: **Feisul Mustapha

---

## [Author Response · Author response to Decision Letter 0]

7 Jul 2021

Dear Editor,

RESPONSE TO REVIEWER/EDITOR COMMENTS

We would like to thank you and the reviewer for the thorough review of our manuscript and the invaluable comments/recommendation offered for improving the overall quality of the paper.

We have addressed all the issues raised. In some cases, we have provided sufficient explanations for why certain information could be included or not included in the study.

We believe the current version of the paper meets the journal’s publication requirements.

We are very grateful for this wonderful opportunity to publish in your highly esteemed journal.

Please find our responses to the issues raised below:

Editor’s Comments:

Comment 1: Line 164-165: How did the authors exclude articles on diabetes and comorbidity? It’s very common for people with diabetes to have one or more comorbidity e.g., hypertension, cardiovascular disease. Would exclusion of these articles affect the results?

Response: We thank the Editor for this useful comment. Actually, we did not exclude articles on diabetes and comorbidities. The articles we excluded were the ones that recruited participants with different chronic diseases (i.e. some having diabetes and others having hypertension, arthritis, cardiovascular disease, etc.). Such studies focused on the generic chronic disease self-management education (CDSME). However, the focus of our study was on diabetes self-management education (DSME), a specific type of CDSME. Thus, it is not that articles with participants having diabetes and other chronic conditions (comorbidity) were excluded. Articles that recruited participants with different chronic conditions, but separated results on diabetes patients from patients with the other chronic diseases were not excluded from the review. Those ones, we were able to extract the information needed for our analysis. [See lines 180-185 on page 7 in the revised manuscript].

Comment 2: Line 170: What was the rationale of excluding articles with 2 or more DSME interventions? Given that the authors aimed to determine the types and effects of DSME interventions, would including these articles, provide more results that are robust? See line 177 on page 7: They compared two or more DSME interventions (e.g., group vs. individual education).

Response: We are grateful for this useful comment seeking a very important clarification. The idea was to look at interventions generally and not papers comparing types of interventions. Based on the mode of delivery, different types of DSME interventions have been distinguished, including, Group Education, Individual Education, and Group & Individual Education Programs. When studies compare only these types of interventions with no controls (i.e. patients with no educational intervention), it is difficult to determine their effectiveness. We have added further information to clarify this (please see Page 8, lines 186-193 of the revised manuscript).

Comment 3: Has this systematic (scoping) review been registered in PROSPERO?

Response: The study has not been registered in PROSPERO 

Comment 5: Line 315: Since there were 14 studies with HbA1c outcome, a meta-analysis should be performed, taking into account the study designs.

Response: We thank the Editor for this comment: Our aim was to describe the nature and types of DSME interventions that have been implemented in the WHO African Region. Also, the studies were not homogeneous enough to conduct a meta-analysis. For, instance, in terms of study design, 11 out of the 19 studies were randomized controlled trials (RCTs) although 14 studies assessed HbA1c. Also, although these 11 studies measured the same outcome (HbA1c), they were so diverse in terms of the subjects involved and the interventions implemented (please see Tables 2 and 3 for descriptions of the subjects and the DSME interventions). Thus, combining these studies that differ substantially in a meta-analysis could yield a meaningless summary result. [Also, see lines 382-391 on page 27 in the discussion for details]

Comment 6: Similar comments applied to the changes in weight, BMI and blood pressure

Response: Same as our response to comment 5

Comment 7: Figure 1: Please provide the reasons of excluding 169 articles.

Response: In the initial submission, the reasons for exclusion were provided. Please see Page 10, lines 245-248 for these reasons. We have also included the information in Figure 1. 

Comment 8: How many articles were solely targeting either type 1 diabetes or type 2 diabetes? Could the outcome assessment be stratified by the type of diabetes? See page 11, Lines 248-251:

Response: Sixteen studies included patients with type 2 diabetes, while only two included participants with both type 1 & type 2 diabetes. One study did not specify the type of diabetes patients who participated in the DSME intervention. No article solely targeted type 1 diabetes (Please see Page 11, lines 264-267 for this information). 

Journal Requirements:

Editor’s comment: We note that this manuscript is a systematic review or meta-analysis; our author guidelines therefore require that you use PRISMA guidance to help improve reporting quality of this type of study. Please upload copies of the completed PRISMA checklist as Supporting Information with a file name “PRISMA checklist”.

Response: We are grateful for this comment. Initially, we submitted the paper as a systematic review. However, after the internal assessment by PLOS ONE, the manuscript was returned to us to change it to a scoping review before it could be sent out for external review. We, thus, changed the paper type from systematic review to scoping review and added the corresponding PRISMA extension for scoping review checklist. Please find below the first editorial review report we received from PLOS ONE Editorial Office:

PONE-D-21-04506

Diabetes self-management education interventions in the WHO African Region: a systematic review Dr Emmanuel Kumah

Dear Dr. Kumah,

We've checked your submission and before we can proceed, we need you to address the following issues:

1. Thank you for your submission to PLOS ONE.

We note that you define your submission as a systematic review. We feel that it is actually a scoping review, a similar type of review that PLOS ONE considers along with other forms of systematic review. Please see general criteria here: https://journals.plos.org/plosone/s/submission-guidelines#loc-systematic-reviews-and-meta-analyses.

We request that authors substitute a completed version of the PRISMA extension for scoping reviews (https://www.equator-network.org/reporting-guidelines/prisma-scr/) for their current PRISMA checklist. Please attach a completed checklist to your submission as supporting information.

In addition, please change all references to "systematic review" to "scoping review" throughout the manuscript and in the full and short titles.

We look forward to hearing from you

We've returned your manuscript to your account. Please resolve these issues and resubmit your manuscript within 21 days. If you need more time, please email the journal office at plosone@plos.org. We are happy to grant extensions of up to one month past this due date. If we don't hear from you within 21 days, we will withdraw your manuscript.

Please log on to PLOS Editorial Manager at https://www.editorialmanager.com/pone/ to access your manuscript. You will find your manuscript in the 'Submissions Sent Back to Author' link under the New Submissions menu. Be sure to remove your previous manuscript file if you are uploading a new file in response to these requests. After you've made the changes requested above, please be sure to view and approve the revised PDF after rebuilding the PDF to complete the resubmission process.

We are requesting these changes to comply with the PLOS ONE submission guidelines (https://journals.plos.org/plosone/s/submission-guidelines). Please note that we won't send your manuscript for review until you have resolved the above requests.

Thank you for submitting your work to PLOS ONE and supporting our mission of Open Science.

Kind regards,

Samantha Russell

PLOS ONE

Reviewer #1

Comment 1: Reference #2 is not appropriate for line 68 as this article is referencing another source. Please reference appropriately

Response: Thank you very much for this important observation. The primary source has been referenced appropriately (please see Reference #2 of the revised manuscript).

Comment 2: Reference #5 is not appropriate for line 73, line 82 & line 86 as this article is referencing other sources. Please reference appropriately. Line 86 is correct (please see page 623 of the article cited)

Response: This has also been corrected (please see Reference # 5 of the revised manuscript). However, the primary source of the information on Line 86 is from Atun et al. (2017) (Reference # 8 of the revised paper). This is contained on Page 623 of their article: Diabetes in sub-Saharan Africa: from clinical care to health policy.

Comment 3: Line 128: is that SI or S1?

Response: Please it is S1 and has been corrected accordingly

Comment 4: Line 170: Could the authors explain the rationale for this exclusion point?

Response: The idea was to look at interventions generally and not papers comparing types of interventions. Also, it is difficult to determine the effectiveness of DSME interventions when studies only compare these interventions with no corresponding control groups (i.e. patients not exposed to any intervention). So we developed our inclusion/exclusion criteria to eliminate any study comparing only DSME interventions without corresponding control groups.

Comment 5: Line 171: could have included qualitative studies or mixed methods – if the outcome is behavioural change – and this would have more depth to the results while still keeping to the study objectives. I also noted you included one 1 mixed methods study (line 240). 

Response: We acknowledge that this is a limitation of our study. We have therefore recommended that future research should focus on qualitative evaluation of DSME in the WHO African Region. [See lines 470-471 on page 31 of the revised manuscript]

Comment 6: Can the authors clarify the difference between ‘clinics’ and ‘health centers’ Line 264-265? In some countries, these are the same (only difference in name)

Response: Clinics and health centers are the same as they both focus on outpatient care. However, within the setting of most countries in the WHO African Region, health centers are community-based health facilities focusing on general primary care, while most clinics are specialized health facilities focusing on specific diseases and conditions. [See Lines 291-295 on page 15 of the revised manuscript]

Comment 7: Please also explain what is a ‘health promoter’ Line 273 (paid or non-paid non-medically trained individuals? Volunteers?)

Response: Health promoters are paid non-medically trained professionals whose work is to promote public health. This information has been added on Page 16, lines 301-302. 

Thank you

---

## [Decision Letter · Decision Letter 1]

2 Aug 2021

Diabetes self-management education interventions in the WHO African Region: a scoping review

PONE-D-21-04506R1

Dear Dr. Kumah,

We’re pleased to inform you that your manuscript has been judged scientifically suitable for publication and will be formally accepted for publication once it meets all outstanding technical requirements.

Kind regards,

Lee-Ling Lim

Academic Editor

PLOS ONE

Additional Editor Comments (optional):

All comments have been adequately addressed. 

Reviewers' comments:

Reviewer's Responses to Questions

**Comments to the Author**

1. If the authors have adequately addressed your comments raised in a previous round of review and you feel that this manuscript is now acceptable for publication, you may indicate that here to bypass the “Comments to the Author” section, enter your conflict of interest statement in the “Confidential to Editor” section, and submit your "Accept" recommendation.

Reviewer #1: All comments have been addressed

2. Is the manuscript technically sound, and do the data support the conclusions?

Reviewer #1: Yes

3. Has the statistical analysis been performed appropriately and rigorously? 

Reviewer #1: Yes

4. Have the authors made all data underlying the findings in their manuscript fully available?

Reviewer #1: Yes

5. Is the manuscript presented in an intelligible fashion and written in standard English?

Reviewer #1: Yes

6. Review Comments to the Author

Reviewer #1: (No Response)

7. PLOS authors have the option to publish the peer review history of their article (what does this mean?). If published, this will include your full peer review and any attached files.

Reviewer #1: **Yes: **Feisul Mustapha

---

## [Editor Report · Acceptance letter]

5 Aug 2021

PONE-D-21-04506R1 

Diabetes self-management education interventions in the WHO African Region: a scoping review 

Dear Dr. Kumah:

I'm pleased to inform you that your manuscript has been deemed suitable for publication in PLOS ONE. Congratulations! Your manuscript is now with our production department. 

Kind regards, 

on behalf of

Dr. Lee-Ling Lim 

Academic Editor

PLOS ONE